# National-scale spatiotemporal patterns of vegetation fire occurrences using MODIS satellite data

**Upenyu Naume Mupfiga**[1,2]*, **Onisimo Mutanga**[1], **Timothy Dube**[3]

**1** Discipline of Geography, School of Agricultural, Earth and Environmental Sciences, University of KwaZulu-Natal, Pietermaritzburg, South Africa, **2** Department of Geography, Environmental Sustainability and Resilience Building, Midlands State University, Gweru, Zimbabwe, **3** Department of Earth Sciences, Institute of Water Studies, The University of the Western Cape, Bellville, South Africa

* nyatondou@gmail.com

**Data Availability Statement:** All relevant data are within the paper and its Supporting information files.

**Funding:** The author(s) received no specific funding for this work.

## Abstract

As the risk of climate change increases, robust fire monitoring methods become critical for fire management purposes. National-scale spatiotemporal patterns of the fires and how they relate to vegetation and environmental conditions are not well understood in Zimbabwe. This paper presents a spatially explicit method combining satellite data and spatial statistics in detecting spatiotemporal patterns of fires in Zimbabwe. The Emerging Hot Spot Analysis method was utilized to detect statistically significant spatiotemporal patterns of fire occurrence between the years 2002 and 2021. Statistical analysis was done to determine the association between the spatiotemporal patterns and some environmental variables such as topography, land cover, land use, ecoregions and precipitation. The highest number of fires occurred in September, coinciding with Zimbabwe's observed fire season. The number of fires significantly varied among seasons, with the hot and dry season (August to October) recording the highest fire counts. Additionally, although June, July and November are not part of the official fire season in Zimbabwe, the fire counts recorded for these months were relatively high. This new information has therefore shown the need for revision of the fire season in Zimbabwe. The northern regions were characterized by persistent, oscillating, diminishing and historical spatiotemporal fire hotspots. Agroecological regions IIa and IIb and the Southern Miombo bushveld ecoregion were the most fire-prone areas. The research findings also revealed new critical information about the spatiotemporal fire patterns in various terrestrial ecoregions, land cover, land use, precipitation and topography and highlighted potential areas for effective fire management strategies.

## 1. Introduction

In semi-arid savannas of Southern Africa, veld fires have maintained the balance between grassy and woody vegetation [1, 2]. While distinct wet and dry seasons in sub-Saharan Africa influence fire activity, the occurrence of fires is affected by several factors including anthropogenic activities and climate variability [3]. Although fire is vital in the existence of savanna

**Competing interests:** The authors have declared that no competing interests exist.

ecosystems, the effect of anthropogenic activities has altered the natural fire regimes, threatening biodiversity [4, 5]. While the release of greenhouse gases (GHG) by forest fires significantly contributes to climate change, the higher temperatures associated with climate change influence the drying of forests, increasing their vulnerability to fires [6]. Air pollution associated with persistent fires poses a health risk to both humans and ecosystems [7, 8].

Due to the recurrent nature of fires in the savanna ecosystems [9], monitoring their spatiotemporal patterns is inevitable for effective fire management. Fire monitoring systems should produce timely and accurate information about the spatiotemporal behaviour of fires. Sustainable management of fire activity requires access to information not only on the location of historical fires but also information on spatiotemporal trends detected early enough to influence future fire trajectory. Remotely sensed data has been useful in fire monitoring [10]. The Moderate Resolution Spectral Radiometer (MODIS) active fire product, for example, has greatly enhanced the analysis of fire occurrence in the landscape by providing spatially detailed, timely and cost-effective information on fire dynamics of global and local importance [11]. The patterns of thousands of individually detected fire data can be difficult to visually evaluate at different spatiotemporal scales. In addition, a critical bottleneck exists in an attempt to unveil the patterns and trends in the available large and complex fire datasets obtained from remote sensing platforms. There is, therefore a need for methods to explore and interpret spatiotemporal patterns of fire data for effective fire management and policy decision-making. Spatiotemporal pattern analysis has the potential to rapidly and accurately identify areas where specific fire management interventions should be prioritized [12]. Spatial statistics greatly assist in the quick identification of spatiotemporal trends of fire activity without the need for pre-existing information on the underlying causal factors [12].

Emerging hotspot analysis is an approach used for statistical spatiotemporal analysis of phenomena. Recently, the approach has been used to detect spatiotemporal patterns of COVID-19 cases [13], landslides [14], invasive species [15] and forest loss [12, 16–18]. A few studies were done in Australia [19] and southeast Asia [5, 20], utilizing the emerging hotspot analysis method to understand the spatiotemporal patterns of fires. In Zimbabwe, limited studies have analyzed the spatial distribution of fire occurrence [21] and its intensity [22] at a national scale. Shekede et al., (2021) assessed the spatial clustering of remotely sensed fire points while Mupfiga et al., (2022) utilized the Getis Ord (Gi*) a spatial autocorrelation method, to detect the spatial hotspots of fire intensity in Zimbabwe. With the few studies done, mainly focusing on the spatial nature of fire occurrence, little is still known about the spatiotemporal patterns of the fires occurring in the study area. Fire management in the study area observes a distinct fire season from 31 July to 31 October, as gazetted in the national statutory instrument of Zimbabwe [23]. It, however, remains unclear whether there have been changes in the fire season over the years.

The occurrence of fires largely depends on the physical characteristics of the fuel load and the vegetation type [24]. The characteristics of vegetation determine the fuel load and its flammability. Succulent vegetation, for example, is less likely to burn due to moisture content and less leaf litter. Topography and land use/land cover are also important factors contributing to fire occurrence [25].

This study utilized a combination of remotely sensed data and spatial statistics, enabling the solving of complex location-based fire monitoring problems [26]. Specifically, the objective of this study was to detect both the spatial and temporal patterns of forest fire occurrence in Zimbabwe. The study utilized both the Getis-Ord (Gi) [27] statistic, which determines spatial clustering, and the Mann-Kendall trend test which detects the temporal trends across the time series [12]. This more robust approach improves on the traditional hotspot analysis methods

which usually analyze the spatial dimension (space-based hotspot analysis) without incorporating the temporal dimension into the hotspot analysis of fire occurrence.

## 2. Materials and methods

### 2.1 Study area

The study was done in Zimbabwe which is located between 15˚30′′ to 22˚30′′ S and 25˚30′′ to 33˚30′′ E in Southern Africa as indicated in Fig 1. The mean sea level of the study area ranges from below 300 m for Southern regions to above 2500 m for the Eastern parts. Zimbabwe is generally characterized by three main seasons, hot and wet (November to April), cool and dry (May to August) and hot and dry (August to November). The mean annual rainfall in Zimbabwe varies from below 400 mm to above 1500 mm [21]. Zimbabwe is classified into seven agroecological zones which are characterized by decreasing mean annual rainfall from agroecological zone 1 (about 1250mm) to zone Vb (below 400mm). Temperatures in the study area increase from agroecological zones I to Vb while elevation decreases. About 95% of the study area's forest cover is covered by savannah woodlands where there is a mixture of trees and grasses which offer fuel for fires. Fire is a major forest ecosystem disturbing factor in Zimbabwe and contributes to deforestation and land degradation [28].

### 2.2 Datasets used for fire occurrence assessment

The MODIS daily active fire data product (MCD14ML) was utilized due to its free accessibility, large area coverage and effectiveness in fire monitoring [25]. The MODIS Terra and Aqua satellite, with a revisit period of 1 to 2 days, detect active fires at 1 km spatial resolution at

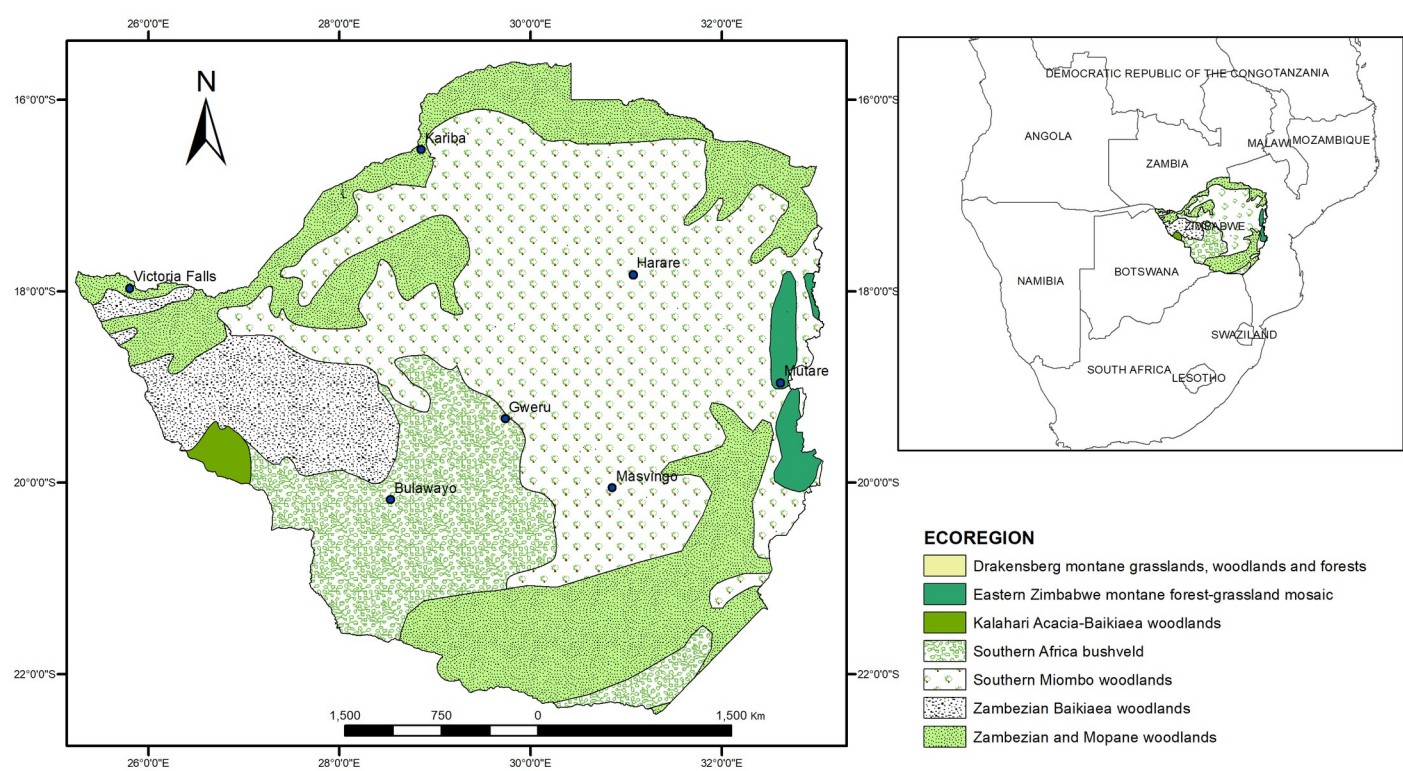

**Fig 1. Map of Zimbabwe showing the ecoregions: Adapted from Olson et al., 2001.**

**Table 1. Datasets used in the study.**

| Data | Source | Resolution |
|---|---|---|
| Modis (MCD14ML) active fire data | https://earthdata.nasa.gov/earth-observation-data/near-real-time/firms/active-fire-data [36] (accessed on 4 March 2023)) | 1 km |
| Land cover | https://viewer.esa-worldcover.org/worldcover/ (accessed on 27 March 2022) | 10m |
| Land use | Environmental Management Agency | N/A |
| Agroecological zones | Zimbabwe National Geospatial and Space Agency (ZINGSA) [37] | N/A |
| Topography | https://earthexplorer.usgs.gov/ (accessed on 15 March 2022) | 30m |
| Precipitation | https://climateknowledgeportal.worldbank.org (accessed on 24 March 2023) | N/A |
| Ecoregions/ biomes | https://www.worldwildlife.org/publications/terrestrial-ecoregions-of-the-world (accessed on 15 March 2023) | N/A |

nadir but can also detect fires smaller than 1 km$^2$. The detection of a fire is dependent on the fire temperature, the angle of the satellite and the prevailing weather conditions during the detection time [29]. Despite the course spatial resolution, fire data from the MODIS sensor was preferred in this study because of the channels on the MODIS sensor that are specifically designed for fire monitoring [30]. In addition, the MODIS sensor has long-term fire records, high temporal resolution as well as high precision of the fire points [31]. The MODIS collection 6 (MODIS C6.1) fire product, used in this study, has been improved to give low commission and omission errors [31].

The fire events (between 2002 and 2021) utilized in the study were downloaded for free from Fire Information Management System (FIRMS) website (https://earthdata.nasa.gov/earth-observation-data/near-real-time/firms/active-fire-data) (accessed on 4 March 2023) as a shapefile (*shp). Attributes associated with the dataset include location (latitude and longitude), date and time of data acquisition, fire radiative power and the confidence level. The acquisition of MODIS data is clearly described in Giglio and Justice (2003). Additionally, based on the fire data attributes, only presumed vegetation fire points were utilized in the data analysis. The variables shown in Table 1 were also utilized in the analysis based on their association with fire occurrence [32, 33].

## 2.3 Satellite data pre-processing

All the data (Table 1) used in this study was projected to the UTM coordinate system in ArcMap 10.5 using the Projection and Transformation tool. To minimize false fire alarms and maximize reliability, only fire points with a confidence level greater than 30%, were utilized in the analysis [34, 35].

## 2.4 Data analysis

**2.4.1 Annual, monthly and seasonal fire trends.** The fire counts were extracted from the fire point map and the total number of fires detected for each year was calculated. To test whether fire data was normally distributed, the Shapiro-Wilk test of normality [38] was performed using the "shapiro.test" function in the R programming software. The normality test results indicated that the fire data utilized in this study did not follow a normal distribution hence non-parametric statistical methods were used. To test whether total annual fire counts were statistically different over the 20-year study period, the Kruskal-Wallis rank sum test, a non-parametric test, was applied to the data, using the "kruskal.test" function in R studio. To determine the correlation between the annual fires and annual rainfall acquired from https://climateknowledgeportal.worldbank.org (accessed on 24 March 2023), the Spearman's rank

correlation method was utilized using the "cor.test()" function in the R studio programming package.

The monthly average fire counts detected in the study area from 2002 to 2021 were calculated. The Kruskal-Wallis rank sum test was used to test whether the monthly fire counts were significantly different over the study period. The "kruskal.test" function in R studio was used for the analysis. The number of fire counts detected during the three seasons (wet and dry, cool and dry, hot and dry) for the whole study period was calculated. To analyze the seasonal variation of fire occurrence in the study area, the Kruskal Wallis rank sum test was also applied to the fire data.

**2.4.2 Emerging hot spot analysis.**   The emerging hotspot analysis was used to identify spatial and temporal trends and patterns of fire occurrence in the study area. The spatial statistic method combines the utilization of the Getis-Ord (Gi*) [27] statistic to determine the location and level of clustering and the Mann-Kendall trend test to evaluate the temporal trends across the time series [12]. The Mann-Kendall test, a rank correlation method, analyses whether there is a decreasing or increasing trend in a given time series data.

One important component of the emerging hotspot analysis method is the space-time cube, which is a descriptive statistic contained in bins, where the geographic location (x and y) is represented by the base of every bin while the height (z) represents time [17, 26]. Before running the emerging hotspot analysis, the Space Time Pattern Mining Tool in a GIS environment was used to create the space-time cube, using the MODIS fire data from 2002 to 2021. This tool utilizes the non-parametric Mann-Kendall [39, 40] trend test to estimate the temporal trends for each fire location. The trend analysis is based on a comparison of the assumed result of having no significant trend over time against the observed result. The trend for each bin is shown by a z-score where a positive and negative z-score indicates an increasing or decreasing trend respectively.

The space-time cube was then used as an input into the emergence hotspot analysis to determine the spatiotemporal pattern of the fires from 2002 to 2021. The Getis-Ord Gi* [27] statistic analyses spatial clustering and determines variability within clusters, assigning z scores for all bins. The neighbourhood distance and neighbourhood time step parameters define how many surrounding bins in space and time are considered during the calculation of the statistic for a specific bin. For this study, the neighbourhood distance was set at 1km and the neighbourhood time step interval was set at one year.

For every feature in the input feature file, the emerging hotspot analysis makes a new output feature class with a z-score, p-value and the significance of the trend is shown by the p-value. A statistically significant hotspot, for example, with a z-score greater than 1.96 and p-value less than 0.05 has a higher clustering intensity. The emerging hotspot analysis results in seventeen categories [12]. In this study, an emerging hotspot refers to locations where the observed spatiotemporal patterns are not due to random, but represent areas where underlying spatiotemporal processes are at play (Getis and Ord 1992). The emerging hotspot analysis method, using the Mann-Kendall statistic, tests whether there exists a significant temporal trend within the 20-year fire data.

**2.4.3 Association between spatiotemporal patterns of fire and agroecological regions.**
The fire spatiotemporal patterns map was overlaid with the agroecological regions map to analyze their association. The agroecological zone map utilized in the analysis was developed based on the spatial distribution of temperature and rainfall conditions in the study area [37]. The association between the spatiotemporal patterns of fire and the agroecological regions was statistically tested using the chi-square test in R programming.

**2.4.4 Association between spatiotemporal patterns of fire and terrestrial ecoregions.**
The occurrence of fires largely depends on the physical characteristics of the fuel load and the

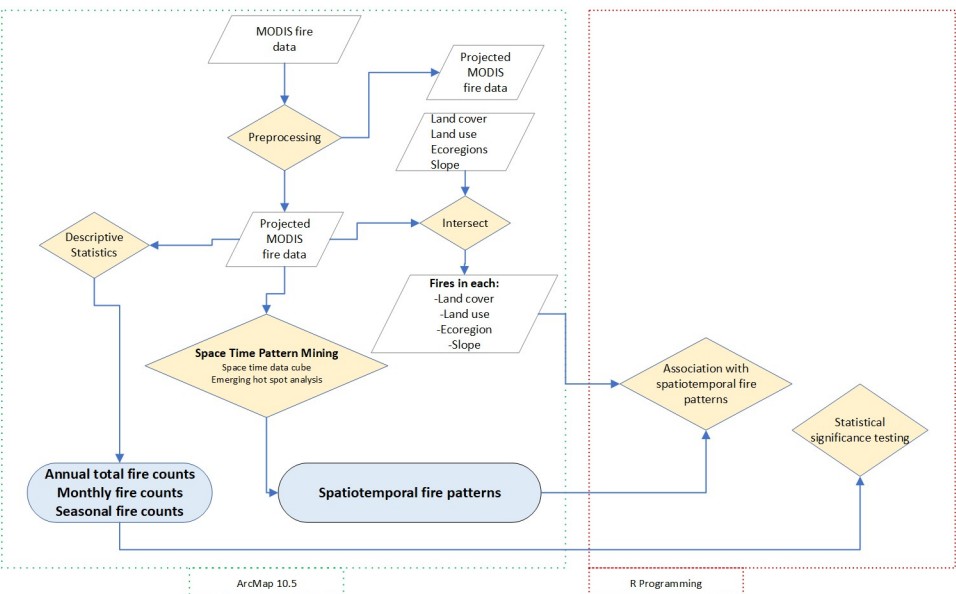

**Fig 2. Outline of the methods for data processing and analysis used in the study.**

vegetation type [24]. Terrestrial ecoregions data from the World Wildlife Fund (WWF) [41] was therefore included in the analysis to determine the association between the fire occurrence pattern and vegetation types. The ecoregions data showing distinct biotas was downloaded from http://www.worldwildlife.org/publications/terrestrial-ecoregions-of-the-world (accessed 15 March 2023). The association between the terrestrial ecoregions and the spatiotemporal fire patterns was analyzed using overlay analysis in a GIS environment and a chi-square test was used to statistically test the significance of the association.

**2.4.5 Association between spatiotemporal patterns of fire and land use and land cover types.** Land cover types derived from the land cover map acquired from https://viewer.esa-worldcover.org/worldcover/ (accessed on 27 March 2022) shown in Table 1 were used in this study to analyze their association between the spatiotemporal fire patterns. The land use and the land cover maps were each overlaid with the spatiotemporal fire patterns in a GIS. The chi-square test was used to analyze the association between the spatiotemporal patterns of fire and the landcover types in the study area.

**2.4.6 Association between spatiotemporal patterns of fire and topography.** The association between the spatiotemporal patterns of fire and topographic variables was analyzed. Topographic variables, slope and aspect were derived from the digital elevation model downloaded from https://earthexplorer.usgs.gov/ (accessed on 15 March 2022). Overlay analysis in a GIS was done to relate topography and the spatiotemporal fire patterns in Zimbabwe. The statistical significance of the association was analyzed using the chi-square test. The methods used in the study are outlined in Fig 2.

## 3. Results

### 3.1 Annual, monthly and seasonal fire trends

The results from the analysis of the MODIS fire data from the study area have shown (Fig 3) that fires were detected every year during the study period. Each box plot in Fig 3a shows the

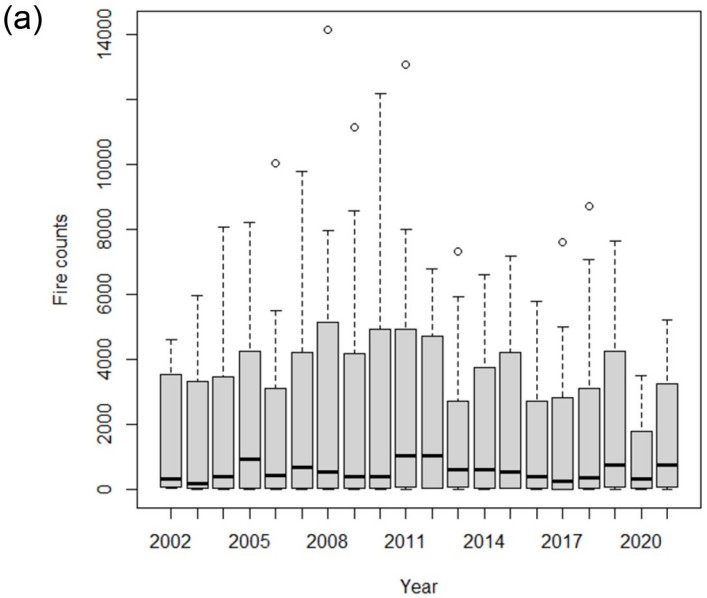

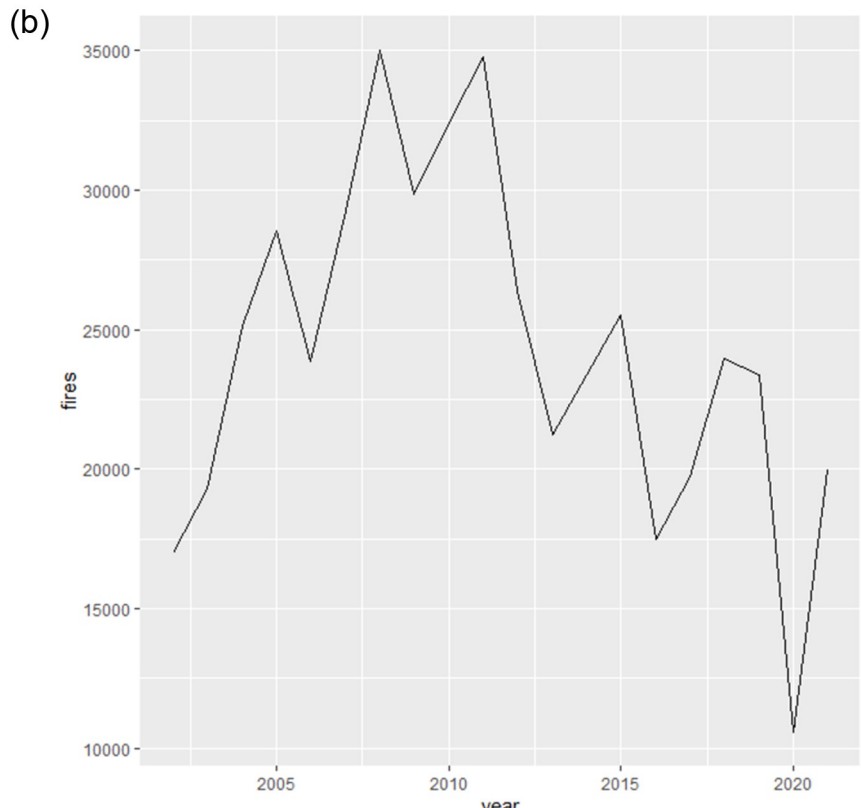

**Fig 3. a**. Boxplots showing the annual distribution of the fire counts from 2002–2021 **b**. Total annual fire trend from 2002–2021.

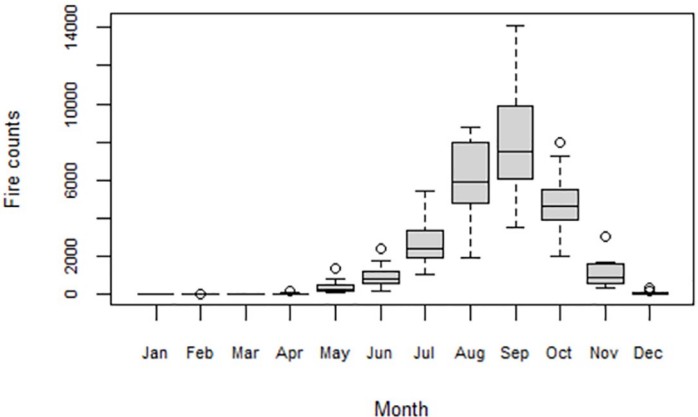

**Fig 4. Boxplots showing the monthly distribution of fire counts from 2002–2021.**

summary of all the fires that occurred in the respective year. The highest total number of fires was detected in 2008, while the lowest total fire incidents occurred in 2020. There was an upward trend in the total annual fire counts from 2002 to 2010, while from 2011 to 2021 there was a general downward trend as clearly shown in Fig 3a and 3b. The Mann-Kendall trend test revealed a downward trend (tau = -0.2) which was, however, not statistically significant ($p > 0.05$). The results of the Spearman's rank correlation test revealed that in the first 10 years of the study period, there was a weak non-significant correlation (rho = 0.224 and $p > 0.05$) between total annual fire occurrence and rainfall. On the other hand, in the last 10 years of the study period, the correlation between total annual fire occurrence and annual rainfall was negative (rho = 0.44) and not statistically significant ($p > 0.05$).

Fig 4 clearly shows the average monthly fire counts over the 20-year study period based on the detection of fires by the MODIS sensor. Each box plot represents a summary of all the fires that occurred in each month over the 20 years. It is evident from Fig 4 that over the study period fire activity was experienced in every month of the year. Fire activity significantly increases from June, with a peak in September, then declines from October as shown in Fig 4.

The higher monthly fire counts coincide with the legally defined fire season which spans from 31 July to 31 October [23] in Zimbabwe. Although June, July and November are not included in the official fire season, the fire activity during these months is relatively high.

The temporal distribution of fire counts based on climatic seasons in Zimbabwe during the study period is clearly illustrated in Fig 5. The hot and dry season is characterized by significantly higher fire activity than the other seasons. During the hot and wet season, the fire counts detected in the study area were generally below 2000. The least number of fires were detected during the cool and dry season. The Kruskal Wallis rank test results showed a significant difference (Table 2) in the number of fire counts among seasons.

## 3.2 Spatiotemporal fire pattern analysis

**3.2.1 Emerging hotspot analysis.** The analysis of the spatiotemporal pattern of fire incidents which occurred in Zimbabwe during the study period is illustrated in Figs 6 and 7 with over 60% of the fires showing an oscillating cold spots pattern. Only 10% of the fires did not show any spatiotemporal pattern. The oscillating fire cold spots are dominant in the central and northwestern parts of the study area. The fire hot spots detected in the northern districts

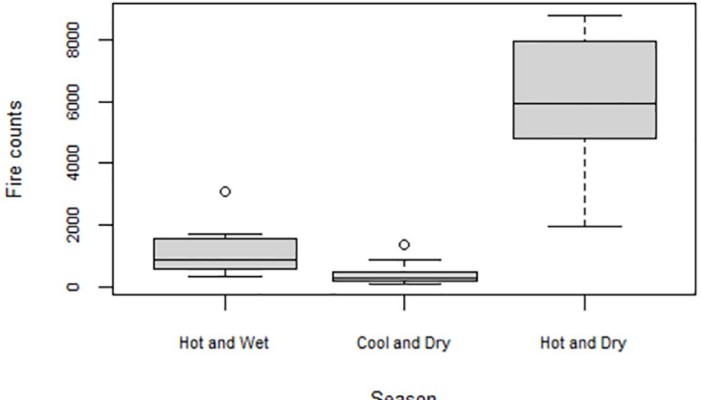

**Fig 5. Boxplots showing the number of fires occurring during the hot and wet (November to April), cool and dry (May to July) and hot and dry (August to October) seasons from 2002 to 2021.**

**Table 2. Kruskal-Wallis rank test results.**

|  | Kruskal-Wallis test statistic | p-value | Significance |
|---|---|---|---|
| Mean monthly fire counts | 219.84 | < 2.2e-16 | ** |
| Mean seasonal fire counts | 46.974 | 6.305e-11 | ** |

NB asterisks * depicts not significant ** depicts significant fires

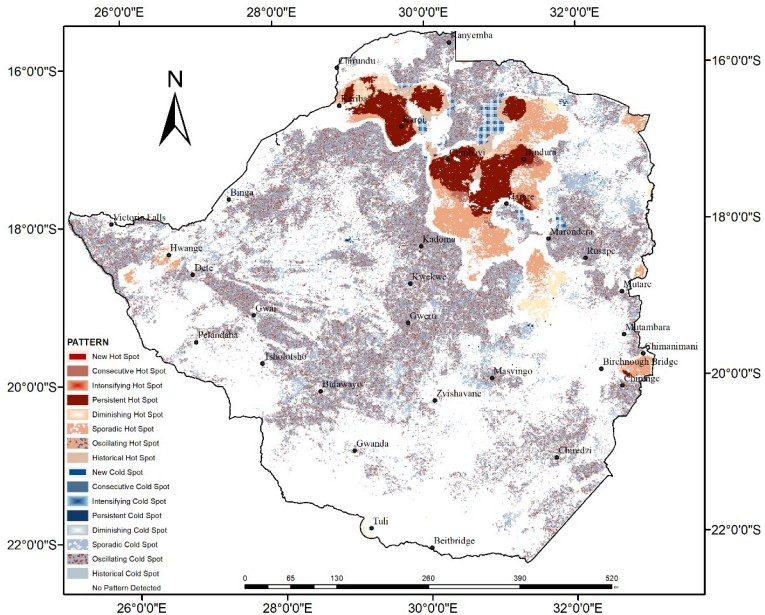

**Fig 6. The spatiotemporal pattern of fire occurrence from 2002 to 2021.**

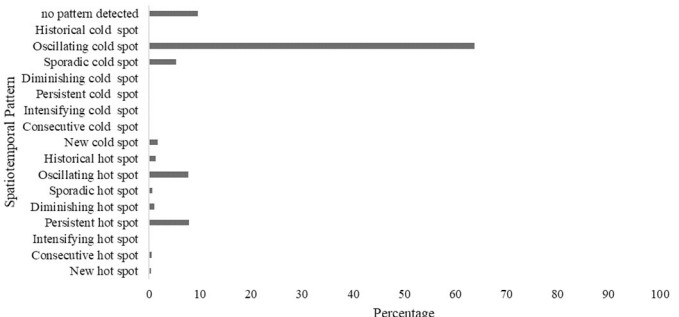

**Fig 7. The proportions of the spatiotemporal fire patterns.**

such as Bindura, Chinhoyi, Kariba, Karoi and Muzarabani, exhibited oscillating, sporadic and persistent patterns.

Interestingly some of the spatiotemporal fire hot spots detected in the northern parts of the study area exhibited historical and diminishing spatiotemporal patterns. A few fire incidents were classified into sporadic fire hot spot pattern in the eastern and western parts of the study area. The proportions of the various spatiotemporal fire patterns detected in the study area are illustrated in Fig 7. Generally, most of the fires detected in Zimbabwe during the study period are characterized by an oscillating pattern.

To give a clear visualization of the distribution of only the spatiotemporal hotspots detected in the study area Fig 8 has been presented. The map clearly shows that besides the distribution of the different patterns of hotspots in the northern parts of the study area, there were patches of oscillating hotspots in the eastern and western parts of the study area specifically in Chipinge and Hwange areas.

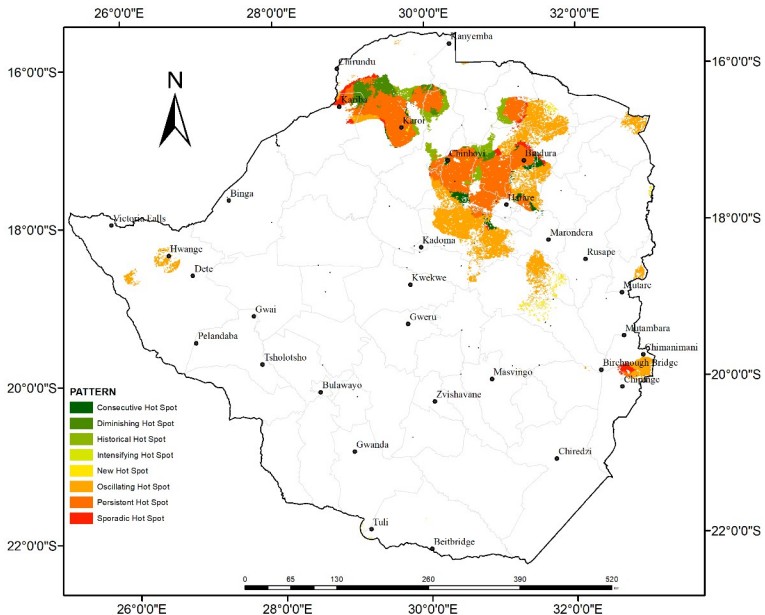

**Fig 8. The spatiotemporal patterns of hot spots only from 2002 to 2019.**

**3.2.2 Association between spatiotemporal fire hotspots with agroecological zones.** The oscillating and the persistent hotspots are the most prevalent patterns of temporal fire clusters in the study area as shown in Fig 9a. Regions IIa and IIb are the most fire-prone areas in the study area. The oscillating spatiotemporal pattern was exhibited by the fire hotspots observed in the agroecological zone I which lies in the eastern mountainous part of Zimbabwe.

Fig 9b clearly shows that the Southern Miombo bushveld is the most fire-prone ecoregion in Zimbabwe as shown by the prevalence of persistent and oscillating hotspots. Historical, diminishing and sporadic spatiotemporal pattern of fires was also observed in the Southern Miombo woodlands. There was also fire activity observed in the Zambezian Mopane woodlands and the Eastern Montane forest ecoregions over the study area as evidenced by the detection of oscillating, persistent and sporadic fire hot spots.

**3.2.3 Association between spatiotemporal patterns of fire and land use types and topography.** The spatiotemporal patterns of the detected fires and their association with the

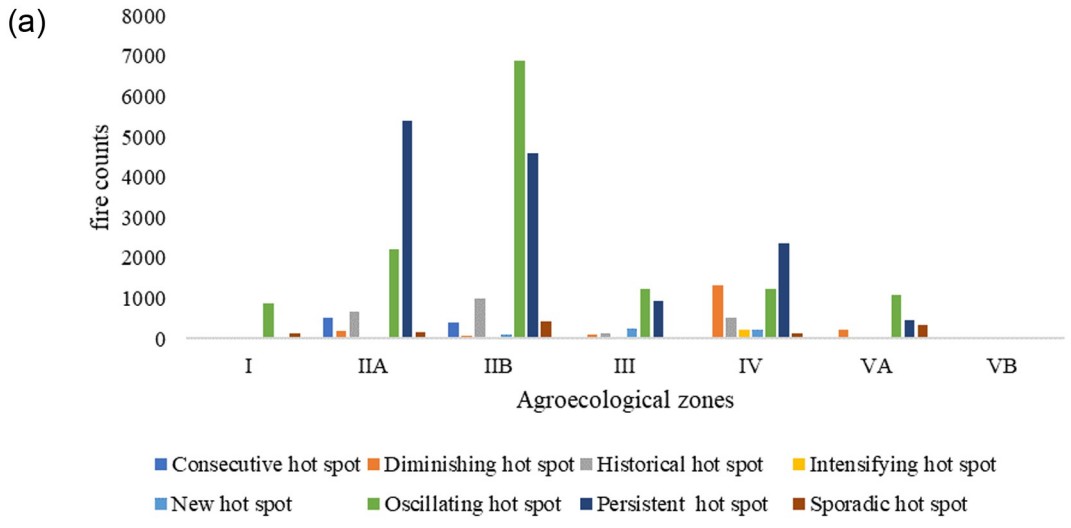

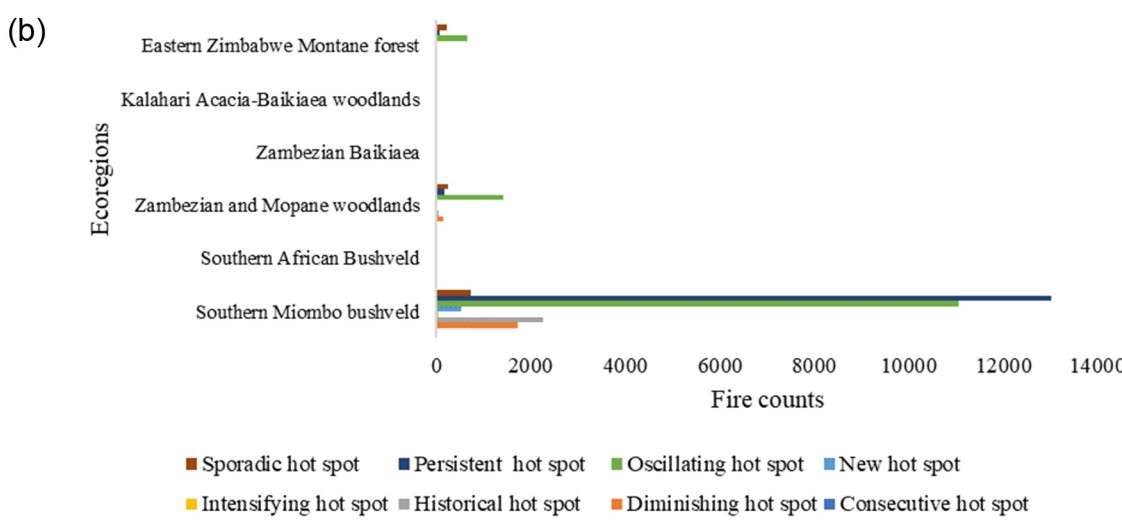

**Fig 9.** Spatiotemporal patterns of fire in **a**) agroecological zones and **b**) terrestrial ecoregions.

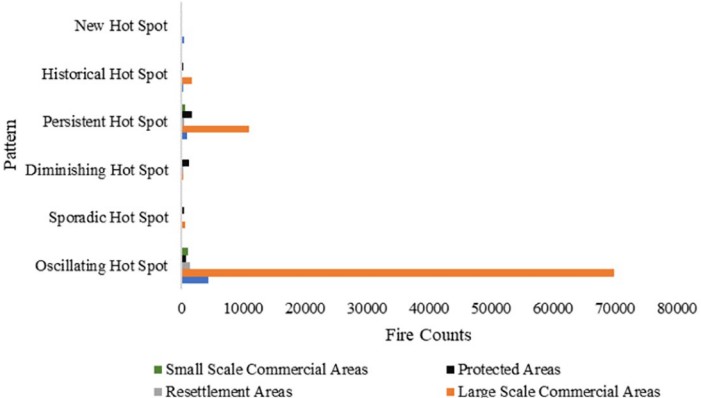

**Fig 10. Association between spatiotemporal patterns of fire and land use types.**

various land use types are shown in Fig 10. It is evident that the large-scale commercial areas were highly affected by fire over the 20-year study period with the oscillating fire hotspot pattern being most prevalent. Persistent and historical fire hot spots were also detected in the study area. Interestingly, there is evidence of persistent, diminishing and oscillating hot spots within protected areas which is highly unexpected. The association between the spatiotemporal patterns of fire occurrence and land use types and topography was statistically significant.

The spatiotemporal pattern of fires and their occurrence in different slope conditions shown in Fig 11 significantly ($p<0.05$) show that most fires in Zimbabwe occur in areas on gentle slope with persistent and oscillating fire hotspots constituting 45% and 28% of the fire hot spots respectively. There was also a small proportion of the persistent and oscillating fire pattern occurring in areas lying on moderately steep slopes. There were no fires detected in very steep areas (>45˚).

## 4. Discussion

The study has assessed the utility of satellite data and emerging hot spot analysis to detect the spatiotemporal patterns of fires that were detected by the MODIS satellite sensor from the years 2000–2019. The general analysis of the average annual fire occurrence data reveals that there was a slight non-significant downward trend (Kendall Tau = -0.2 and $p>0.05$) in the fire

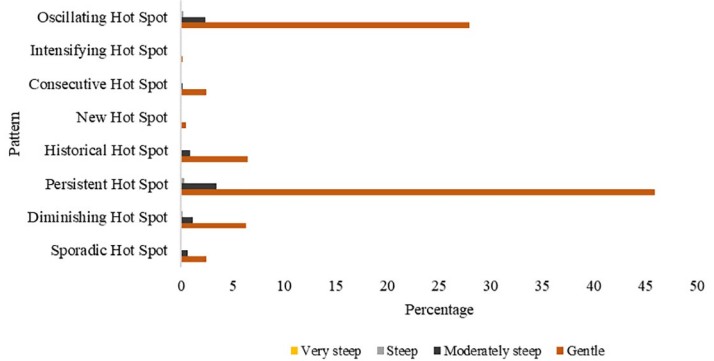

**Fig 11. Association between spatiotemporal patterns of fire and slope.**

activity in the study area over the 20-year study period. The study findings clearly show an upward trend in the occurrence of fires from 2002 to around 2010 while a downward trend in fire occurrence was observed from 2010 to 2021. The upward trend in fire occurrence observed in this study however agrees with findings by [42] who observed an increasing trend in fire occurrence in one district of Zimbabwe from 2001 to 2009. The research findings however contrast with findings from Shekede et al., 2021 where an increasing trend in fire counts was observed over Zimbabwe but within a different period (2000 to 2019). This variation could be due to the differing start and end times of the fire data analysis. The downward trend observed in this study from 2010 to 2010 could be associated with improved fire management strategies being implemented by responsible authorities.

This study has shown that high fire activity predominantly occurs during the dry and hot season. The seasonality of fire occurrence in Zimbabwe is influenced by the country's two distinct rainfall-related seasonal patterns, the wet season from November to March and the dry season from April to October [37]. During the wet season, for example, vegetation in Zimbabwe grows due to the abundant rainfall, and the susceptibility to fire occurrence is generally low. The dry season coincides with the fire season in Zimbabwe which is associated with the drying up of vegetation due to high temperatures and lack of rainfall and becomes highly combustible. Fire occurrence is significantly (Table 2) high between July and September, which is the driest and windiest of the year. The number of fire incidents tends to decline towards the end of the dry season as the first rains begin to fall, and the vegetation becomes moister and less prone to fires.

The research findings have also shown that although June, July and November are not included in the official fire season, the fire counts recorded for these months are relatively high. This newly unveiled information is critical for fire management decisions which may lead to considering the shifting of the fire season in Zimbabwe. This information from the study findings strengthens the perception highlighted in [43] to change the fire season dates and the amendment of statutory instrument provision. A detailed study on the analysis of possible shifts in fire seasonal occurrence over a long period may be required. Understanding the seasonality of fire occurrence is crucial for the development of timely and effective fire management strategies in Zimbabwe.

The study has revealed that, based on the analysis of the 20-year MODIS fire occurrence data, the spatiotemporal patterns vary from oscillating cold spots in the southern and central parts to persistent, oscillating, diminishing and historic hot spots in the northern parts of the study area. Persistent hotspots pattern refers to the occurrence of fire in all the time steps [12]. The sporadic hot / coldspots, for example, represent areas where fires disappear and reappear over time while diminishing hot or cold spots are those areas which have become less of a fire hotspot over the years [19]. The occurrence of diminishing and historical spatiotemporal fire patterns could be associated with changes in the extent and quality of vegetation in these areas. The reduction in vegetation cover results in reduced fuel available for burning hence the potential reduction in fire occurrence. New hotspots are areas which were never hotspots until the final time step which is the 2021 fire season [12]. These emerging spatiotemporal fire patterns are expected to guide the allocation of resources for fire monitoring and management by responsible authorities in Zimbabwe. Local investigations could also be done in areas where persistent hot spots were observed to determine the drivers of the spatiotemporal pattern. The spatiotemporal fire pattern analysis therefore assists in identifying priority areas for conservation [12].

The research findings unveil new information on the patterns of temporal fire clusters in the various ecoregions indicating the risk that fires pose on the ecoregions within the study area. The persistent and oscillating fire hot spot patterns were associated with Southern

Miombo Bushveld which has shown to be prone to fire. Ecoregions are characterized by different vegetation and climatic conditions which affect the fire occurrence patterns [44]. Miombo woodlands are generally susceptible to fire and burn mainly during the dry season. The occurrence of the persistent fire hot spot in the Southern Miombo Bushveld ecoregion is not surprising because the biome is highly associated with persistent fires [45]. The information on the association between ecoregions and spatiotemporal fire clusters supports the development of effective fire management policies which benefit economic, social and ecological objectives.

Agroecological zones IIa and IIb, characterized by moderate rainfall and high temperatures, are the most fire-prone regions with oscillating and persistent hotspots being more prevalent. This concurs with literature which indicates the high occurrence of fires in such regions [22, 28]. The oscillating fire hotspots detected in the eastern region (Agroecological zone I) could be associated with burning occurring in forest plantations. While higher elevation may be associated with less probability of fire occurrence due to cool and moist conditions, local factors can influence this relationship. The higher fire activity within the large-scale commercial farms could be associated with the Fast Track Land Reform Program where burning was a common practice as the farmers prepared their new land [42, 46].

The research findings have clearly shown a significant association between fire occurrence and gentle slopes. This can be highly associated with human contribution to fire occurrence where most agricultural activities are done in flat areas [33]. Most farmers utilize fire during land preparation in Zimbabwe [42, 46].

This study has shown the utility of remote sensing in assessing a past fire regime, which is useful in fire management policy formulation. There could however be limitations on the MODIS fire detection where cloud cover, smoke or dense forest canopies could have obscured the detection of some fires [47]. With a spatial resolution of 1km, the MODIS sensor could also have a chance of missing too small fires.

Ground truthing for validation was not possible because of the historical nature of the fire data used in the analysis. The detailed process of validating MODIS active fire data has, however, been provided by [48]. To improve the observational accuracy of the data, the techniques used by the Fire Information Resource Management System (FIRMS) to validate the active fire data are provided by [49].

Although this study analyzed how terrestrial ecoregions, agroecological zones and topography are associated with fire occurrence, the causes of fire occurrence were not within the scope of this study as remote sensing methods cannot assess such information. Although the temporal scale of this study was restricted by the availability of the MODIS sensor, the timescale used is adequate to study a fire regime of the study area [47].

The research findings from this study provide valuable information for the evaluation of the fire management policies and plans implemented in the Zimbabwe over the study period. The research findings also add new information for decision-making regarding resource allocation for fire management. The study findings can also assist in identifying areas where incentives should be provided to the responsible communities for improvement in fire management. With the rise in climate warming and population, it is important to understand the long-term spatiotemporal patterns in fire occurrence [2, 9].

This study focused on the whole country of Zimbabwe and resulted in a valuable but coarse map of the spatiotemporal fire patterns throughout the study area. The causes and drivers of the spatiotemporal patterns such as persistent and diminishing hotspots, which were out of this study's scope, should be investigated in future studies. The contribution of climate change, land use changes and human activities on the spatiotemporal distribution of fires in the study area could also be explored in future studies.

## 5. Conclusion

This paper has unveiled the utility of the emerging hotspot analysis detecting the spatiotemporal patterns of fire occurrence in Zimbabwe using MODIS fire data over 20 years. The research findings have clearly shown that based on MODIS fire data and emerging hot spot analysis, significant spatiotemporal patterns of fires were detected. Additionally, fire occurrence in Zimbabwe is strongly seasonal, with the highest frequency of fires occurring during the dry season from April to October. The spatiotemporal fire pattern in Zimbabwe is related to variations in vegetation types and topography. Overall, the study has shown the utility of the emerging hot spot analysis method as a robust tool in detecting spatiotemporal patterns of fires. The results hence provide timely information to fire management decision-makers. Informed fire management strategies such as resource allocation can therefore be implemented. The revision of the current fire season could also improve the effectiveness of the fire management strategies in the study area.

## Supporting information

**S1 Data.**
(7Z)

## Acknowledgments

We acknowledge the use of data from NASA's Fire Information for Resource Management System (FIRMS) (https://earthdata.nasa.gov/firms), part of NASA's Earth Observing System Data and Information System (EOSDIS).

We also acknowledge the use of the global terrestrial ecoregions data from https://www.worldwildlife.org/publications/terrestrial-ecoregions-of-the-world which was provided freely.

## Author Contributions

**Conceptualization:** Upenyu Naume Mupfiga.

**Data curation:** Upenyu Naume Mupfiga.

**Formal analysis:** Upenyu Naume Mupfiga.

**Investigation:** Upenyu Naume Mupfiga.

**Methodology:** Upenyu Naume Mupfiga, Onisimo Mutanga, Timothy Dube.

**Software:** Upenyu Naume Mupfiga.

**Supervision:** Onisimo Mutanga, Timothy Dube.

**Visualization:** Upenyu Naume Mupfiga.

**Writing – original draft:** Upenyu Naume Mupfiga.

**Writing – review & editing:** Upenyu Naume Mupfiga, Onisimo Mutanga, Timothy Dube.

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
