## [Decision Letter · Decision Letter 0]

28 Sep 2023

PONE-D-23-23794National-scale spatiotemporal patterns of vegetation fire occurrences using MODIS satellite dataPLOS ONE

Dear Dr. Mupfiga,

Thank you for submitting your manuscript to PLOS ONE. After careful consideration, we feel that it has merit but does not fully meet PLOS ONE’s publication criteria as it currently stands. Therefore, we invite you to submit a revised version of the manuscript that addresses the points raised during the review process.

We look forward to receiving your revised manuscript.

Kind regards,

Bhogendra Mishra

Academic Editor

PLOS ONE

Journal Requirements:

3. We note that Figures 1, 6 and 8 in your submission contain map images which may be copyrighted. All PLOS content is published under the Creative Commons Attribution License (CC BY 4.0), which means that the manuscript, images, and Supporting Information files will be freely available online, and any third party is permitted to access, download, copy, distribute, and use these materials in any way, even commercially, with proper attribution. For these reasons, we cannot publish previously copyrighted maps or satellite images created using proprietary data, such as Google software (Google Maps, Street View, and Earth). For more information, see our copyright guidelines: http://journals.plos.org/plosone/s/licenses-and-copyright.

1.) You may seek permission from the original copyright holder of Figures 1, 6 and 8 to publish the content specifically under the CC BY 4.0 license.  

2.) If you are unable to obtain permission from the original copyright holder to publish these figures under the CC BY 4.0 license or if the copyright holder’s requirements are incompatible with the CC BY 4.0 license, please either i) remove the figure or ii) supply a replacement figure that complies with the CC BY 4.0 license. Please check copyright information on all replacement figures and update the figure caption with source information. If applicable, please specify in the figure caption text when a figure is similar but not identical to the original image and is therefore for illustrative purposes only.

**Additional Editor Comments:**

Reviewers' comments:

Reviewer's Responses to Questions

**Comments to the Author**

1. Is the manuscript technically sound, and do the data support the conclusions?

Reviewer #1: Yes

Reviewer #2: Yes

2. Has the statistical analysis been performed appropriately and rigorously? 

Reviewer #1: Yes

Reviewer #2: Yes

3. Have the authors made all data underlying the findings in their manuscript fully available?

Reviewer #1: Yes

Reviewer #2: Yes

4. Is the manuscript presented in an intelligible fashion and written in standard English?

Reviewer #1: Yes

Reviewer #2: Yes

5. Review Comments to the Author

Reviewer #1: 

I have now commented on “National-scale spatiotemporal patterns of vegetation fire occurrences using MODIS satellite data”. You will see that there are a number of issues that need to be addressed before the paper can be accepted for publication by Plos One.

The paper falls within the general scope of the journal. The title reflects the content, abstract is good, and the keywords need to be revised (It is recommended not to include keywords in the title).

In the introduction, the objectives of the article should be stated more precisely.

In general, the description of methods is appropriate. During the past 22 years, has the scope of these ecoregions not changed? I think it is better to use an updated map (Fig 1).

In general, the description of results is appropriate.

Language overall is good.

I would recommend MINOR revision for the paper for it to be considered further for publication.

Reviewer #2: The paper well written and very novel. However the authors are encouraged to attend to the following; Line 82-87 should not sound like a research methodology but should clearly state the main and specific objective (s) of the study.

6. PLOS authors have the option to publish the peer review history of their article (what does this mean?). If published, this will include your full peer review and any attached files.

Reviewer #1: No

Reviewer #2: **Yes: **Pedzisai Kowe

---

## [Author Response · Author response to Decision Letter 0]

15 Dec 2023

The responses to the reviewers have been attached.

---

## [Editor Report · Decision Letter 1]

3 Jan 2024

National-scale spatiotemporal patterns of vegetation fire occurrences using MODIS satellite data

PONE-D-23-23794R1

Dear Dr. Mupfiga,

We’re pleased to inform you that your manuscript has been judged scientifically suitable for publication and will be formally accepted for publication once it meets all outstanding technical requirements.

Kind regards,

Bhogendra Mishra

Academic Editor

PLOS ONE

---

## [Editor Report · Acceptance letter]

19 Jan 2024

PONE-D-23-23794R1 

PLOS ONE

Dear Dr. Mupfiga, 

I'm pleased to inform you that your manuscript has been deemed suitable for publication in PLOS ONE. Congratulations! Your manuscript is now being handed over to our production team.

Kind regards, 

on behalf of

Dr Bhogendra Mishra 

Academic Editor

PLOS ONE